# Lipoprotein(a) and Blood Monocytes as Factors for Progression of Carotid Atherosclerosis in Patients with Premature Coronary Heart Disease

**DOI:** 10.3390/diseases13070196

**Published:** 2025-06-26

**Authors:** Alexandra V. Tyurina, Olga I. Afanasieva, Marat V. Ezhov, Elena A. Klesareva, Tatiana V. Balakhonova, Sergei N. Pokrovsky

**Affiliations:** 1A.L. Myasnikov Institute of Clinical Cardiology, National Medical Research Center of Cardiology n.a. acad. E.I. Chazov, Ministry of Health of the Russian Federation, Moscow 121552, Russia; tvbdoc@gmail.com; 2Institute of Experimental Cardiology, National Medical Research Center of Cardiology n.a. acad. E.I. Chazov, Ministry of Health of the Russian Federation, Moscow 121552, Russia; afanasieva.cardio@yandex.ru (O.I.A.); eaklesareva@cardio.ru (E.A.K.); dr.pokrovsky@mail.ru (S.N.P.)

**Keywords:** lipoprotein(a), immune cells blood count, carotid atherosclerosis

## Abstract

Background. Elevated lipoprotein(a) [Lp(a)] levels are a key factor in the early formation and progression of atherosclerosis. Monocytes in individuals with an elevated Lp(a) level are represented by an activated inflammatory phenotype and have an increased ability for transendothelial migration. This work studies the association between Lp(a), monocytes, and the progression of carotid atherosclerosis in patients with premature coronary heart disease (CHD). Methods. This study included 102 patients with CHD manifested before 55 in men and 60 in women who underwent two carotid duplex scans with an interval of 5 [3; 8] years. The criteria for the progression of carotid atherosclerosis were the appearance of new plaque and an increase in stenosis by >10% in any of the six segments. The lipid profile, Lp(a), and hematology with the calculation of the lymphocyte–monocyte ratio (LMR) were determined in all the patients. Results. The median blood monocyte count was 0.54 × 10^9^/L, and the median LMR was 4.18. In 70 patients, we revealed the criteria for carotid atherosclerosis progression. The groups did not differ by demographics, risk factors, or the blood lipid and lipoprotein levels, except for Lp(a); this concentration was higher in the patients with carotid atherosclerosis progression. The odds of atherosclerosis progression were highest in the patients with an elevated Lp(a) level and a blood monocyte count above the median (16.8, 3.4–83.0, *p* < 0.001). Carotid atherosclerosis progression was associated with LMR < 4.18 and an elevated Lp(a) level (OR = 4.3, 1.1–17.2, *p* = 0.04) and not associated with the patients with Lp(a) levels < 30 mg/dL and an LMR above the median. Conclusions. An elevated Lp(a) level and monocyte count provide the highest probability of the progression of carotid atherosclerosis in patients with premature CHD.

## 1. Introduction

Carotid atherosclerosis is responsible for at least 30% of ischemic strokes [1]. Inflammation plays a key role in atherogenesis [2]. The proinflammatory properties of lipoprotein(a) [Lp(a)] may be related to the structural features of both the apo(a) protein itself, which has an epitope of binding to oxidized phospholipids, and the lipids comprising it. This leads to the increased secretion of cytokines and the expression of adhesion molecules by endothelial cells and is accompanied by the increased transendothelial migration of monocytes [3,4]. According to several studies, Lp(a) is a carrier of a major pool of oxidized phospholipids [5]. The oxidized phospholipids within Lp(a) are recognized by innate immune cells via toll-like receptors and CD14 and CD36 differentiation clusters as damage-associated molecular patterns (DAMPs), initiating an inflammatory response [6]. The blood cell count is a simple routine test to assess the presence of chronic inflammation. Several studies have shown the significance of the lymphocyte–monocyte ratio (LMR) as a marker of chronic inflammation in coronary and peripheral arterial atherosclerosis [7,8]. The relationship between inflammation and the risk of atherosclerosis progression is further supported by studies demonstrating that the attenuation of an inflammatory response is accompanied by an improvement in the surrogate markers of arterial function [9]. We hypothesized that blood monocytes, as cells involved in atherogenesis, alongside elevated Lp(a) concentrations may contribute to the progression of atherosclerosis in carotid arteries. The aim of our study was to investigate whether the combination of elevated Lp(a) and monocytes counts is associated with the progression of carotid atherosclerosis in patients with the early manifestation of CHD.

## 2. Materials and Methods

This retrospective study included patients consecutively examined in an atherosclerosis department between 2019 and 2021. A total of 102 patients with the manifestation of CHD before the age of 55 years in men and 60 years in women who had undergone carotid duplex scanning twice since the manifestation of CHD with an interval of 5 [3; 8] years were selected (Figure 1). Patients with severe comorbidities and conditions affecting prognosis (dementia, alcohol abuse, and autoimmune and infectious diseases), as well as those on PCSK9 inhibitor therapy or lipoprotein apheresis were not included in this study.

Carotid duplex scanning was performed using an iU-22 and Epic-5 ultrasound system (Phillips, Amsterdam, The Netherlands) equipped with a 3–9 MHz linear array transducer. The common carotid artery, the internal carotid artery, and carotid bifurcation on both sides were scanned for atherosclerotic plaque (6 segments in total) in the anterior, lateral, and posterior planes. According to the Mannheim Carotid Intima-Media and Plaque Consensus [10], carotid atherosclerotic plaque was defined as a focal structure protruding into the arterial lumen by at least 0.5 mm or at least 50% compared with the thickness of the adjacent intima–media or as the focal thickening of the intima–media > 1.5 mm. The progression of carotid atherosclerosis was detected when one of the following criteria was present: the appearance of new plaque or an increase in stenosis of more than 10% in any of the segments [11]. All the patients were examined and interviewed to assess the course of CHD and the presence of risk factors for atherosclerosis. This study was conducted in accordance with good clinical practice and the Declaration of Helsinki. The study protocol was approved by the local ethics committee (No. 251, 25 November 2019), and written informed consent was obtained from all the participating patients before they were included in this study.

Lipid concentrations (total cholesterol (TC), triglycerides (TG), and high-density lipoprotein cholesterol (HDL-C)) were determined by the enzymatic colorimetric method using Hitachi 912 (Roche Diagnostics, Basel, Switzerland) and Architect C-8000 (Abbott, Abbott Park, IL, USA) biochemical analyzers. Quality control was performed using PrecinOshm and Precipath control sera (Roche Diagnostics, Basel, Switzerland). The Lp(a) concentration was determined by enzyme-linked immunosorbent assay using monospecific ram polyclonal antibodies against human Lp(a) as previously described [12]. LDL-C was calculated with the Friedewald equation.

In addition, we calculated the level of corrected LDL-C (LDL-C_corr_), which takes into account Lp(a) cholesterol:

LDL-C_corr_ = LDL-C−0.3 × (Lp(a))/38.7, mmol/L [13], where Lp(a) is the concentration of Lp(a) in mg/dL.

The hematology procedure was performed for all patients; the lymphocyte–monocyte ratio (LMR) was calculated as the ratio of the absolute lymphocyte count to the absolute monocyte count.

Statistical analysis was performed using MedCalc 20.022. (MedCalc Software Ltd., Ostend, Belgium). Variables with a normal distribution are presented as mean ± standard deviation (M ± SD); parameters with a non-normal distribution are presented as medians and the 25th and 75th percentiles. The Kolmogorov–Smirnov test was used to determine distribution normality. The parametric Student’s *t*-test and nonparametric Mann–Whitney test were used to compare the quantitative data of the two groups; Fisher’s exact test was used to evaluate the frequency data between the groups. Spearman’s rank correlation, multivariate analysis, and logistic regression were conducted. The odds ratio (OR) and 95% confidence intervals (CIs) were calculated to assess the relationships between the study parameters and the outcomes.

## 3. Results

The progression of carotid atherosclerosis was detected in 70 (69%) of the 102 patients; in 36 of them (35%), new atherosclerotic plaque appeared in previously unaffected segments. In 17 patients (26%), the degree of at least one case of carotid artery stenosis increased by 10%; in 12 (18%), by 11–20%; in 29 (44%), by 21–30%; and in 8 (12%), by 31% or more. The patients with and without the progression of carotid atherosclerosis did not differ in age, the frequency of classical risk factors, or the achievement of target LDL-C and LDL-Ccorr levels during therapy (Table 1).

The groups also did not differ in the number or composition of blood leukocytes according to hematology at the time of repeat carotid duplex scanning (Table 2).

The combination of a monocyte count below the median and an Lp(a) level ≥ 30 mg/dL was found in 35% of the patients with the progression of carotid atherosclerosis versus 10% of the patients without progression, *p* < 0.01 (Figure 2). The progression of carotid atherosclerosis was associated with a decreased LMR combined with an increased Lp(a) concentration (OR 4.3; 1.1–17.2, *p* = 0.04) and not with the patients with Lp(a) levels < 30 mg/dL and LMR values below the median. The groups were comparable in the lipid profile parameters both at the baseline, i.e., before lipid-lowering therapy initiation, and at the time of repeat carotid duplex scanning, except for the Lp(a) concentration (Table 3).

The progression of carotid atherosclerosis was associated with a blood monocyte count above the median against a background of elevated levels of TC, non-HDL-C and Lp(a), but not LDL-C and LDL-C corr. The patients with Lp(a) levels ≥ 30 mg/dL and a blood monocyte count above the median had the highest probability of carotid atherosclerosis progression, OR 16.8 (95% CI 3.4–83.0), *p* < 0.001 (Table 4).

According to logistic regression analysis, Lp(a) concentration ≥ 30 mg/dL ((OR 4.5; 1.6–12.5), *p* < 0.01) and a monocyte count above the median ((OR 2.8; 1.0–7.6), *p* < 0.05) were associated with the progression of carotid atherosclerosis regardless of sex, statin therapy, and the attainment of LDL-C levels < 1.4 mmol/L.

## 4. Discussion

Premature CHD affects a population of patients who, until they have an ischemic event, cannot be classified as a high-risk group in accordance with the existing guidelines for the stratification of cardiovascular risk. At the same time, patients with premature CHD represent a group of concern due to the unfavorable long-term outcome and the high probability of a recurrent ischemic event, myocardial infarction, and ischemic stroke [14]. One out of five patients with premature CHD dies during 10 years of follow-up [15]. The results of our previous study showed that an increased monocyte count as well as an elevated Lp(a) level are associated with a 2.7-fold increase in the probability of MACE and not with those with Lp(a) < 30 mg/dL and a monocyte count < 0.54 × 10^9^/L [14]. The main finding of this study is that the patients with the concomitant elevation of the Lp(a) level and the monocyte count had the highest probability of carotid atherosclerosis progression.

By single factor analysis, the patients with an elevated Lp(a) concentration and blood monocyte counts above the median had the highest probability of the progression of carotid atherosclerosis, OR 16.8 (95% CI 3.4–83.0), *p* < 0.001. In logistic regression analysis, an Lp(a) concentration ≥ 30 mg/dL (OR 4.5 (1.6–12.5), *p* < 0.01) and a blood monocyte count above the median (2.8 (1.0–7.6), *p* < 0.01) were associated with progression of carotid atherosclerosis regardless of sex, statin therapy, or the attainment of LDL-C levels < 1.4 mmol/L.

The pathogenesis and progression of atherosclerotic lesions is a complex process in which the cells of innate immunity play an important role.

According to the lipid hypothesis of atherosclerosis, the penetration and retention of atherogenic lipoproteins into the vascular intima is a critical event initiating an inflammatory reaction in the vessel wall and the development of atherosclerosis. Endothelial dysfunction and monocyte infiltration into the subendothelial space are the key events of atherogenesis [16].

The study of the structure, chemical composition, and proinflammatory properties of lipoprotein particles isolated from plaque extracted during endarterectomy from carotid arteries demonstrates the presence of apoB-containing lipoproteins. In turn, three-dimension electron microscopy of plaque revealed areas of the co-localization of extracellular lipoprotein particles, foam cells, and cholesterol crystals [17].

The previous studies have shown that the monocyte count has a strong positive correlation with the risk of CHD and plays a crucial role in the progression of coronary atherosclerosis [18]. There is also evidence that an elevated monocyte count is an independent risk factor for ischemic stroke [19].

Lp(a) concentration is one of the most significant factors of residual risk. Despite this study, the basis of such high-level atherogenicity of Lp(a) and its influence on the risk of cardiovascular events is still not fully understood. However, the influence of Lp(a) on immune cells as one of the possible and key mechanisms of the initiation and maintenance of chronic inflammation in the vascular wall becomes more evident [20]. Monocytes and macrophages are the first line of defense in innate immunity [21], and their interaction with Lp(a) is actively studied [22,23]. Using morphological and immunohistochemical approaches, Lp(a) has been identified in human atherosclerotic plaque and localized in areas rich in macrophage cells [24].

According to the study of Stiekema L and colleagues, monocytes from healthy subjects and those with CHD and elevated Lp(a) concentrations were characterized by a more pronounced proinflammatory gene expression profile. Additionally, a recent study showed that the lipidome of healthy individuals with elevated Lp(a) levels was also characterized by elevated levels of diacylglycerol and lysophosphatidic acid. Diacylglycerols, in turn, can induce the proinflammatory phenotype of monocytes by changing the signature of the corresponding genes in primary human monocytes and leading to their activation, which is accompanied by the secretion of cytokines (interleukins 6, 8, and 1β) and increased transendothelial migration [3].

The ability of Lp(a) to stimulate the production of proinflammatory monocytes at the bone marrow level by hematopoietic cells and progenitor cells [25] confirms the involvement of the innate immune system in atherogenesis in patients with an elevated Lp(a) concentration.

At present, it has been convincingly established that in persons with high level of Lp(a) against a background of already existing traditional risk factors, the overall global risk is significantly underestimated. The effect of chronic inflammation on the cardiovascular risk in subjects with elevated Lp(a) concentrations has been actively studied in recent years, but the results remain controversial [26].

The LMR as a simple clinical biomarker of inflammation and immune response combines lymphocytes and monocytes into a single index. Our study demonstrated that combination of a reduced LMR and Lp(a) ≥ 30 mg/dL is associated with the progression of carotid atherosclerosis with an OR of 4.3 (1.1–17.2), *p* = 0.04, but this is not the case for the patients with normal Lp(a) levels and an LMR above the median. In addition, the combination of a reduced LMR and an elevated Lp(a) level was significantly more common in the patients with the progression of carotid atherosclerosis. A low LMR is known to be associated with inflammation and oxidative stress [27]. In earlier studies, a decreased LMR was an independent risk factor for coronary heart disease [28], inversely correlated with the severity of coronary atherosclerosis [29] and unfavorable outcomes in patients with stenosing peripheral arterial atherosclerosis.

Increased monocyte content or a decreased LMR against the background of an Lp(a) concentration ≥ 30 mg/dL may be a convenient marker for predicting the progression of carotid atherosclerosis in patients with the early manifestation of CHD. The constancy of the Lp(a) concentration and its ability to influence the cells involved in atherogenesis, with the involvement of various pathogenetic mechanisms, necessitates the determination of the Lp(a) concentration not only in secondary, but also in primary prevention settings.

## 5. Study Limitations

Although power analysis confirms the adequacy of the sample size for primary endpoint assessment, the interpretation of the results for the subgroups (stratified by monocyte concentration relative to the median) should be performed with caution due to the limited sample size within these strata.Carotid artery ultrasound imaging was performed in accordance with the Mannheim consensus guidelines. However, this method is operator-dependent. Currently, there are no universally accepted criteria for defining carotid atherosclerosis progression. Based on a comprehensive literature review, we defined progression as either (1) the development of new atherosclerotic plaque in a previously unaffected arterial segment or (2) an increase in stenosis severity by ≥10%. These criteria were selected to minimize inter-operator variability.The currently available therapeutic options for lipoprotein(a) correction remain limited, and the lymphocyte–monocyte ratio lacks widespread clinical application in routine practice. While our findings demonstrate both scientific and potential practical significance, they require further validation before implementation in clinical practice.

## 6. Conclusions

A concomitantly elevated Lp(a) level and monocyte count provide the highest probability of the progression of carotid atherosclerosis in patients with premature CHD.

## Figures and Tables

**Figure 1 diseases-13-00196-f001:**
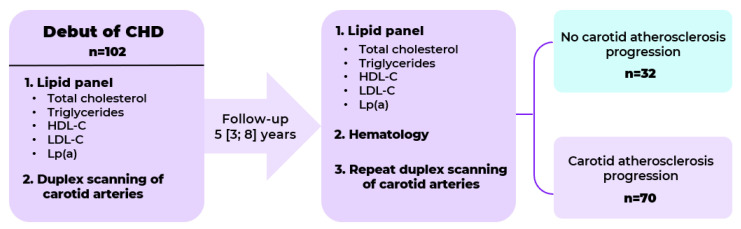
Study design.

**Figure 2 diseases-13-00196-f002:**
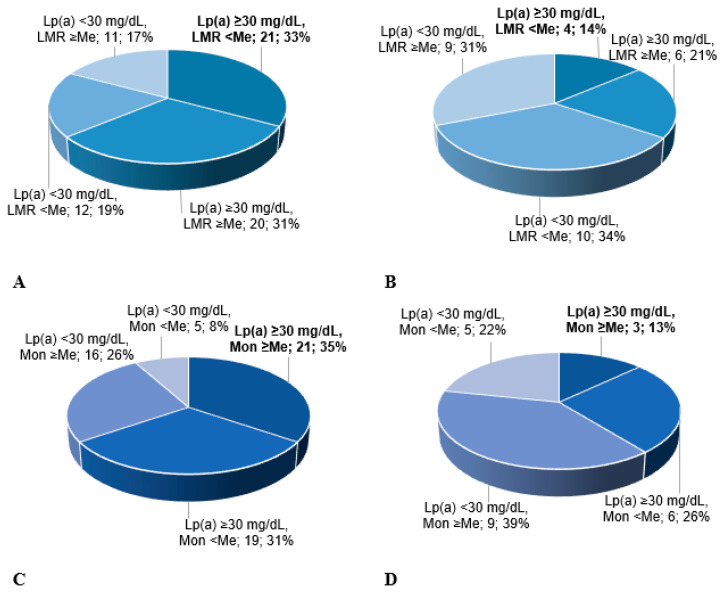
Distribution of patients with (**A**,**C**) or without (**B**,**D**) carotid atherosclerosis progression according to Lp(a) concentration, lymphocyte–monocyte ratio (**A**,**B**), and absolute monocyte count (**C**,**D**). LMR—lymphocyte–monocyte ratio. Me—median. Median for LMR = 4.18. For monocytes, this is 0.54 × 109/L.

**Table 1 diseases-13-00196-t001:** Patients’ characteristics depending on progression of carotid atherosclerosis.

	Progression of Atherosclerosis		*p*-Value
	Yes *n* = 70	No *n* = 32	
Men	54 (77)	24 (75)	0.8
Age, years	63 ± 9.7	59 ± 10.1	0.06
Age at CHD debut, years	50.3 ± 10.3	53.3 ± 10.8	0.2
Follow-up, years	5.5 ± 2.7	4.1 ± 2.1	0.1
Obesity	34 (49)	20 (63)	0.2
Hypertension	56 (80)	29 (91)	0.3
Blood pressure normalization	13 (19)	12 (33)	0.08
Current smoking	12 (17)	4 (13)	0.8
Former smoking	32 (46)	15 (47)	1.0
Type 2 diabetes mellitus	21 (30)	11 (34)	0.7
Statin therapy	62 (89)	23 (72)	0.02
LDL-C < 1.4 mmol/L	9 (13)	4 (13)	1.0
LDL-C_corr_ < 1.4 mmol/L	28 (40)	9 (28)	0.3

Data are presented as M ± SD and *n* (%).

**Table 2 diseases-13-00196-t002:** Hematology parameters in patients at study inclusion depending on further progression of carotid atherosclerosis.

Parameter	Progression of Carotid Atherosclerosis		*p*
	Yes *n* = 70	No *n* = 32	
Leukocytes, 10^9^/L	7.7 [6.5; 8.8]	7.8 [5.9; 8.9]	0.5
Lymphocytes, 10^9^/L %	2.1 [1.6; 2.7]	2.0 [1.7; 2.6]	0.8
	28.2 [24.3; 34.0]	27.6 [25.2; 33.8]	1.0
Neutrophils, 10^9^/L %	4.4 [3.8; 5.5]	4.8 [3.6; 5.4]	1
	61.0 [55.9; 65.1]	60.2 [56.8; 65.6]	0.8
Monocytes, 10^9^/L %	0.55 [0.44; 0.68]	0.50 [0.38; 0.61]	0.1
	7.1 [5.9; 8.7]	6.5 [5.8; 7.6]	0.3
Basophils, 10^9^/L %	0.07 [0.05; 0.100]	0.06 [0.06; 0.09]	0.9
	0.9 [0.6; 1.1]	0.9 [0.7; 1.1]	0.2
Eosinophils, 10^9^/L %	0.14 [0.09; 0.22]	0.13 [0.10; 0.23]	1.0
	1.8 [1.1; 3.4]	2.0 [1.3; 3.0]	1.0
Lymphocyte–monocyte ratio	4.1 [3.0; 4.8]	4.3 [3.4; 5.6]	0.4

**Table 3 diseases-13-00196-t003:** Blood lipids in patients with and without progression of carotid atherosclerosis.

Parameter	At the Debut of CHD		At the Time of Inclusion of Patients	
	Progression		Progression	
	Yes *n* = 70	No *n* = 32	Yes *n* = 70	No *n* = 32
TC, mmol/L	6.3 ± 1.8	6.1 ± 1.7	4.5 ± 1.3	4.5 ± 1.1
Triglycerides, mmol/L	1.5 [1.2–2.2]	1.5 [1.0–1.8]	1.6 [1.1–2.2]	1.6 [1.1–1.8]
HDL-C, mmol/L	1.2 ± 0.3	1.2 ± 0.3	1.1 [1.0–1.36]	1.1 [1.0–1.4]
LDL-C, mmol/L	4.1 ± 1.4	3.9 ± 1.4	2.4 ± 1.0	2.4 ± 0.9
LDL-C_corr_, mmol/L	3.4 ± 1.3	3.6 ± 1.3	1.8 ± 1.1	2.2 ± 0.9
Non-HDL-C, mmol/L	5.15 ± 1.8	4.8 ± 1.7	3.2 ± 1.2	3.2 ± 1.1
Lp(a), mg/dL	70 [19; 118] *	17 [10; 57]	69 [18; 116] *	16 [9; 53]
Lp(a) ≥ 30 mg/dL	43 (61) *	11 (34)	43 (61) *	11 (34)

Data are presented as M ± SD, median [25%; 75%] percentile, and *n* (%), * *p* < 0.05.

**Table 4 diseases-13-00196-t004:** The odds ratio for the progression of carotid atherosclerosis depending on the concentration of atherogenic lipoproteins and the blood monocyte levels.

OR (95% CI)		
	LDL-C < 1.4 mmol/L	LDL-C ≥ 1.4 mmol/L
Monocytes < 0.54 × 10^9^/L	1	1.1 (0.2–7.5)
Monocytes ≥ 0.54 × 10^9^/L	2.0 (0.2–22.0)	1.9 (0.3–12.7)
	LDL-C_corr_ < 1.4 mmol/L	LDL-C_corr_ ≥ 1.4 mmol/L
Monocytes < 0.54 × 10^9^/L	1	0.6 (0.2–2.2)
Monocytes ≥ 0.54 × 10^9^/L	1.9 (0.4–8.8)	1.3 (0.3–4.7)
	Non-HDL-C < 2.2 mmol/L	Non-HDL-C ≥ 2.2 mmol/L
Monocytes < 0.54 × 10^9^/L	1	2.1 (0.6–7.0)
Monocytes ≥ 0.54 × 10^9^/L	1.9 (0.5–7.1)	4.3 (1.1–16.3) *
	TC < 4.0 mmol/L	TC ≥ 4.0 mmol/L
Monocytes < 0.54 × 10^9^/L	1	4.4 (1.2–16.0) *
Monocytes ≥ 0.54 × 10^9^/L	2.8 (0.8–10.3)	3.5 (1.0–11.6) *
	Lp(a) < 30 mg/dL	Lp(a) ≥ 30 mg/dL
Monocytes < 0.54 × 10^9^/L	1	7.6 (1.9–30.5) *
Monocytes ≥ 0.54 × 10^9^/L	4.3 (1.1–16.0) *	16.8 (3.4–83.0) *

Non-HDL-C—non-high-density lipoprotein cholesterol; LDL-C—low-density lipoprotein cholesterol; LDL-_Ccorr_—low-density lipoprotein cholesterol corrected for Lp(a). * *p* < 0.05 relative to the subgroup with monocyte levels below the median distribution and target atherogenic lipoprotein levels achieved.

## Data Availability

The data presented in this study are available upon request from the corresponding author.

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
