# Peer review of "Lipoprotein(a) and Blood Monocytes as Factors for Progression of Carotid Atherosclerosis in Patients with Premature Coronary Heart Disease"

_diseases, 2025, doi:10.3390/diseases13070196_

Round 1
Reviewer 1 Report
Comments and Suggestions for Authors
The manuscript reports the results of a retrospective analysis which tested the predictive value of the composite of serum lipoprotein (a) levels and lymphocyte-monocyte ratio on progression of carotid atherosclerosis. The study is interesting and well presented. I have some questions to propose regarding the methodological aspects:
#1. It is not reported how many patients have been excluded from the study during the 2-year follow-up (2019-2021).
#2. Authors should clarify which was the indication to test serum levels of lipoprotein (a) in 2019-2021, since previous sets of international guidelines on cardiovascular prevention or dyslipidaemias did not recommend this analysis.
#3. Please clarify at what interval carotid duplex scanning was performed.
Author Response
Dear Reviewer,
We thank you for thouroughful evaluation our manuscript and provide pur repy to all your points in the attached file and below. Sincerely, Marat Ezhov
#1. It is not reported how many patients have been excluded from the study during the 2-year follow-up (2019-2021)
Dear Reviewer, thank you for the important comment. Our study is a retrospective analysis of data obtained from patients who had already undergone duplex scanning of carotid arteries twice with an interval of 5 [3; 8] years. Thus, only those 102 patients who met the inclusion criteria (coronary artery disease before age 55 in men and before age 60 in women) and completed both ultrasound examinations were included in the final analysis.
#2. Authors should clarify which was the indication to test serum levels of lipoprotein (a) in 2019-2021, since previous sets of international guidelines on cardiovascular prevention or dyslipidaemias did not recommend this analysis
We thank you for this question. Indeed, international guidelines for cardiovascular disease prevention during that period had not yet included routine lipoprotein(a) determination. However, at our center, lipoprotein(a) levels were measured in many patients for many years as a main part of our research work.
#3. Please clarify at what interval carotid duplex scanning was performed
Thank you for raising this point. Each patient underwent two duplex scanning with median interval of 5 [3; 8] years. The mean follow-up time was: 5.5±2.7 years for patients with atherosclerosis progression (n=70) and 4.1±2.1 years for those without progression (n=32).

Reviewer 2 Report
Comments and Suggestions for Authors
This is a nice paper publishable in a solid general or atherosclerosis, lipid or vascular journal. The limitation includes that there is not much to do about Lpa and nothing for LMR , which is not used by anyone clinically, which should be mentioned . Very important negatives are nothing on physical activity (Sanchis-Gomar F et al. Can J Cardiol 2024 on-line; Martinez-Gomez D etal Prog Cardiovasc Dis 2024;83: 116-123; Carbanas-Sanchez V et al. Mayo Clin Proc 2024; 99: 564-577; Lavie CJ et al . PCVD 2025; 89: 61-62) or cardiorespiratory fitness (Kaminsky LA e al. PCVD 2024; 83: 3-9; Ross R et al. PCVD 2024; 10-15;Sparks JR et al. MCP2024; 99: 1261-1270; Lavie CJ et al. MCP 2025; 100: 402-404), which are both very important for atherosclerosis , CVD and prognosis.
Author Response
This is a nice paper publishable in a solid general or atherosclerosis, lipid or vascular journal. The limitation includes that there is not much to do about Lpa and nothing for LMR , which is not used by anyone clinically, which should be mentioned . Very important negatives are nothing on physical activity (Sanchis-Gomar F et al. Can J Cardiol 2024 online; Martinez-Gomez D metal Prog Cardiovasc Dis 2024;83: 116-123; Carbonates-Sanchez V et al. Mayo Clin Proc 2024; 99: 564-577; Lavie CJ et al . PCVD 2025; 89: 61-62) or cardiorespiratory fitness (Kaminsky LA e al. PCVD 2024; 83: 3-9; Ross R et al. PCVD 2024; 10-15;Sparks JR et al. MCP2024; 99: 1261-1270; Lavie CJ et al. MCP 2025; 100: 402-404), which are both very important for atherosclerosis , CVD and prognosis.
Dear Reviewer, we thank you for your comments. We fully agree with you regarding the necessity of discussing of the limited therapeutic options for lipoprotein(a) and the clinical applicability of the lymphocyte-monocyte ratio. In the discussion, we acknowledge the existing limitations and perspectives in therapeutic approaches for correcting Lp(a) levels. In the final version of the manuscript, we will add to the "Study Limitations" section an indication that our results are both of scientific and practical significance and require further validation for clinical application.
Physical activity and cardiorespiratory fitness are indeed significant modifiable risk factors for ASCVD, as demonstrated in the referenced studies, and may influence the progression of carotid atherosclerosis. However, within the scope of this study, physical activity status was not assessed and, consequently, not evaluated. Therefore, we do not consider it justified to expand the reference list with the citations you have suggested.